mSystems

ᵃ | **Open Peer Review** | Host-Microbial Interactions | Research Article

# Revealing pathogenesis-associated metabolites in *Histoplasma capsulatum* through comprehensive metabolic profiling

Adrian Heckart,[1] Jean-Christophe Cocuron,[2] Stephanie C. Ray,[3] Chad A. Rappleye,[3] Ana P. Alonso[1,2]

**ABSTRACT** During infection, *Histoplasma capsulatum* yeasts interact with a variety of phagocytic cells, where macrophages represent an important niche for long-term intracellular fungal survival and replication. In the phagosomes of macrophages, a hostile environment where most microorganisms are killed, *Histoplasma* not only survives but overcomes several biological challenges and proliferates intracellularly. To better understand the characteristics of intracellular *Histoplasma* and the phagosomal environment, a metabolomic platform was used to analyze *Histoplasma* yeasts cultured on different carbon sources and yeasts extracted from macrophages, identifying metabolites associated with pathogenesis. Metabolomic results of *in vitro*-grown yeasts were further characterized with available transcriptomic data, informing underlying gene expression patterns in response to contrasting milieus. These approaches revealed that *Histoplasma* yeasts, unlike many other yeasts, do not ferment sugars to ethanol, and, when cultivated on glycolytic versus gluconeogenic carbon sources, produce distinct metabolomes with altered intracellular amino acid, lipid, and sugar contents. Furthermore, analysis of *Histoplasma*-inoculated media illustrated that *Histoplasma* secretes mannitol and anthranilates. Lastly, a comparison of the metabolomes derived from *in vitro* cultivation versus intracellular growth highlighted leucine and cysteine/cystine as amino acids, which may serve as sources of carbon, nitrogen, and sulfur to yeasts within macrophages. These results detail metabolites linked to *Histoplasma* metabolism during macrophage infection, identifying potential candidates to target for novel histoplasmosis therapeutics.

**IMPORTANCE** Intracellular pathogens reside within host cells, surviving against innate immune responses while exploiting host resources to proliferate. Understanding the mechanisms that underlie their survival and proliferation is critical for developing novel treatments and therapeutics for the diseases these pathogens cause. While *Histoplasma* is a unique example of a true intra-phagosomal pathogen, insights into its pathogenesis may still inform the study of other intracellular pathogens.

**KEYWORDS** *Histoplasma*, host–pathogen, histoplasmosis, metabolomics, mass spectrometry, fungal pathogen

*H*istoplasma capsulatum is a dimorphic human fungal pathogen responsible for the respiratory disease, histoplasmosis. The fungus grows as a mycelial form in soil containing bat or bird guano, and aerosolization of the mycelia-produced conidia and their inhalation causes infections of humans (1). Once inhaled, human body temperature triggers the transformation of the conidia into the pathogenic yeast form (2), where yeasts evade and counteract innate immune defenses (3), infect phagocytic cells, including alveolar macrophages (4, 5), and reside intracellularly within the phagosomal compartment.

**Editor** Ákos T. Kovács, Universiteit Leiden, Leiden, the Netherlands

Address correspondence to Ana P. Alonso, Anapaula.Alonso@unt.edu.

The authors declare no conflict of interest.

See the funding table on p. 17.

10.1128/msystems.00186-25 **1**

To successfully survive and proliferate within macrophage phagosomes, *Histoplasma* must navigate numerous biological obstacles. One significant challenge is nutritional immunity, which allows immune cells to deprive pathogens of essential nutrients, including metals, carbon, nitrogen, and sulfur (6–8). Additionally, *Histoplasma* must neutralize host-derived superoxide generated by the phagocyte NADPH oxidase and other reactive oxygen species produced from superoxide (9). The acidic environment within the phagosome poses another obstacle, activating hydrolytic enzymes that can degrade phagosomal contents (10, 11). Each of these host defenses necessitates that *Histoplasma* deploy adaptive strategies for intracellular survival and growth. For instance, one way it acquires iron is by extracting it from transferrin through modulation of phagosomal pH or by producing hydroxamate siderophores that sequester iron (12–14). The reduction of ferric iron into its usable form, ferrous iron, has also been reported to occur with γ-glutamyl transferase (15). To detoxify host-derived reactive oxygen species, *Histoplasma* produces extracellular superoxide dismutase and an extracellular catalase (CatB) (16, 17).

Despite advances in understanding these defenses, the mechanisms by which *Histoplasma* circumvents nutritional immunity remain the least understood. *In vitro* growth of *Histoplasma* demonstrates it can synthesize most monomers of macromolecules *de novo* from basic carbon and nitrogen sources; however, *Histoplasma* must acquire organic sulfur and thiamine from the phagosomal environment (18–21). For carbon and energy, intracellular metabolic sources have been generally identified as gluconeogenic molecules rather than glycolytic ones (22–24). The proliferation of amino acid auxotrophic mutants within macrophages has shown that phenylalanine and tyrosine are available to *Histoplasma* within the macrophage phagosome, but tryptophan is not (25). Together, these data indicate that the phagosome environment is a distinct niche and not a reflection of the macrophage cytosol nor the glucose-rich medium, in which the macrophages are cultured. Further investigation into other gluconeogenic molecules potentially available in the phagosome is still needed.

To identify metabolites related to *Histoplasma capsulatum* survival and proliferation within the macrophage phagosome, the metabolome of *Histoplasma* grown on different carbon sources was analyzed and compared with the metabolome of *Histoplasma* grown inside macrophages. Additionally, *in vitro* transcriptomic data were compared with the metabolite analysis to infer any gene expression patterns underlying the observed metabolic changes. This approach revealed specific metabolites that may contribute to the ability of *Histoplasma* to overcome nutritional immunity and neutralize reactive oxygen species. These findings open new research avenues into *Histoplasma* pathogenesis and highlight critical metabolic reactions and pathways that could be targeted for histoplasmosis therapeutics.

## RESULTS

### Different growth conditions result in distinct metabolomes

To establish *in vitro* metabolomes for comparative analyses, *Histoplasma* yeasts were cultivated *in vitro* using media with glucose (Glc) or amino acids (AA) as carbon sources, as well as a 1:1 mix of both (Glc + AA). These growth conditions were chosen based on prior studies, which indicated gluconeogenesis is essential for *Histoplasma* growth in the macrophage phagosome, pointing to the metabolism of gluconeogenic rather than glycolytic substrates (22). Gas chromatography–mass spectrometry (GC–MS) and high-resolution liquid chromatography (LC)–MS/MS with two separate chromatographic columns, reverse phase (RP) and hydrophilic interaction liquid chromatography (HILIC), were utilized to maximize the metabolome coverage. Metabolites were extracted using two methods: chloroform/methanol/water (CMW) phase extraction and boiling water (BW) extraction. A comparison of CMW and BW metabolomes was conducted with GC–MS and both HILIC ion mode data sets (Table S1a through f) to determine the most effective procedure for metabolite extraction. CMW extracted a total of 434 annotated features detected with GC–MS and LC–MS/MS in comparison to 381 using BW.

Additionally, CMW was more efficient in extracting AA, organic acids, and secondary metabolites (Fig. S1). Consequently, CMW data sets were selected for statistical analysis (Table S1b, g through j). In total, 627 metabolites across three acquisition methods (GC–MS, LC–MS/MS RP, and LC–MS/MS HILIC) were tentatively identified (Table S1k). Principal component analysis (PCA) revealed distinct metabolomes for Glc, Glc + AA, and AA cultivation (Fig. S2) with tight clustering of quality controls (QC) in all data sets, indicating minimal variance and high reproducibility. Principal component (PC) 1 explained 40.0–47.5% of the variance and separated growth of *Histoplasma* in both AA-containing media from media lacking AA.

Pair-wise analysis of the top five metabolites by positive and negative $\log_2$ fold-change (FC) between each of the three growth conditions showed that Glc supplementation increases sugar metabolites in *Histoplasma* yeasts, while AA supplementation elevates nitrogen-containing compounds (Fig. 1A through C). Comparing growth on AA versus Glc (Fig. 1A), tryptamine, urea, and cystathionine were among the top five metabolites enriched by growth in AA (304-, 202-, and 113-fold, respectively). Similarly, comparing growth in Glc + AA versus Glc (Fig. 1B), urea and acetyl-serine were the most differentially produced (183- and 114-fold, respectively). In either AA-containing conditions (AA or Glc + AA), the relative abundance of two lipids was significantly elevated compared to growth on Glc: lysodiacylglyceryl-3-O-carboxyhydroxymethylcholine 16:4 and ether-linked digalactosyldiacylglycerol 15:1_28:7. Conversely, sugars and other metabolite classes accumulated more during growth on Glc than on AA or Glc versus Glc + AA (Fig. 1A through C), including mannose 1-phosphate, trehalose, turanose, and xylitol. 2-Isopropylmalate, an intermediate in leucine biosynthesis, was also enriched in yeasts when cultivated on Glc alone. This pattern was also observed for glutamylthreonine, a dipeptide, and a molecule with the same mass as the flavonoid, sakuranetin. Finally, aromatic AA derivatives like tryptamine and indoleacetate, as well as short-chain fatty acid carnitines accumulated with growth on AA compared to Glc + AA (Fig. 1C). Altogether, these results demonstrate that cultivation on different carbon sources causes significant shifts in the *Histoplasma* metabolome, affecting AA, lipids, and sugars.

## *Histoplasma* secretes metabolites into its extracellular environment

To determine if *Histoplasma* yeasts modulate their extracellular environment via the consumption or production of metabolites, GC–MS liquid analysis of the media before and after 68 h inoculation was performed (Table S2a). In parallel, volatile organic compounds were monitored by headspace GC–MS. *Histoplasma* yeasts showed no significant ethanol or acetate fermentation (data not shown). Excluding components of the initial growth media, 59 new metabolites were identified (Table S2b), 17 of which were at least 10-fold higher in intensity after inoculation in at least one condition from background (Fig. 2A). Among these, six were aromatic AA derivatives, including phenylacetate, anthranilate, and indoleacetate. Expectedly, these metabolites exhibited the highest intensities in media with either Glc + AA or AA supplementation. Branched-chain AA derivatives and sugar alcohols were also significantly secreted. Specifically, 2-isopropylmalate, a precursor to leucine biosynthesis, was significantly more secreted when Glc was present in the media, whereas beta-alanine, a catabolic product of branched-chain AA, was more abundant when AA were present in the media. Interestingly, mannitol, a sugar alcohol, was abundantly secreted following AA-based growth rather than growth with Glc alone (Fig. 2A).

The AA composition of the AA or Glc + AA media was also analyzed after 68 h of yeast growth to measure AA consumption. Expectedly, many AAs were significantly consumed, particularly alanine, glutamine, isoleucine, leucine, tyrosine, and valine (Fig. 2B). Notably, glutamine was secreted in media with Glc but consumed when AAs were present. Serine was also secreted in both AA-containing media, with higher secretion in the presence of Glc. Aspartate and glycine were not significantly consumed, while methionine, phenylalanine, and threonine were significantly consumed in the Glc + AA

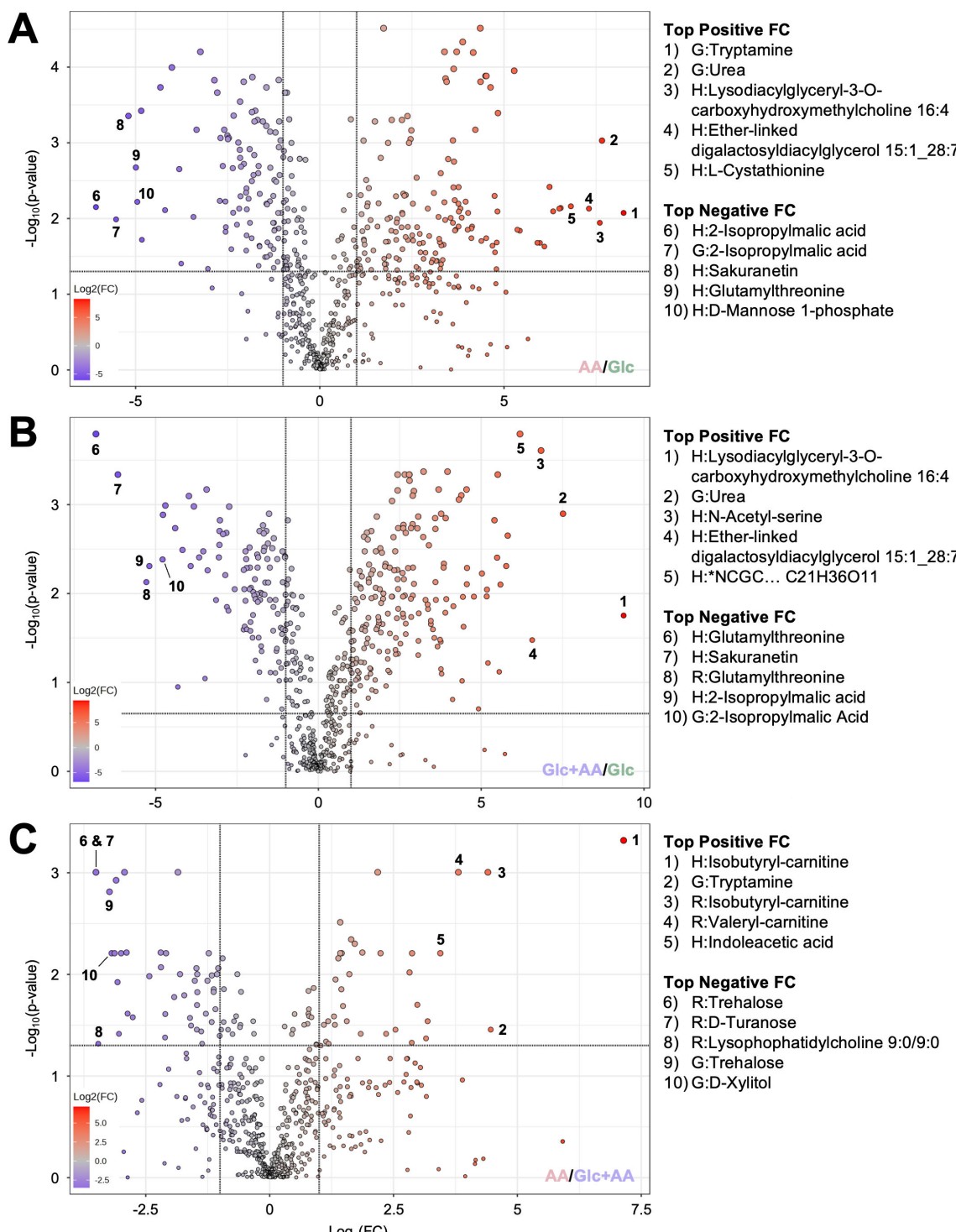

**FIG 1** Glucose (Glc) supplementation increases sugar content, while amino acid (AA) supplementation elevates nitrogen-containing compounds. Volcano plots represent the 627 annotated metabolites via GC–MS (G), HILIC LC–MS/MS (H), and RP LC–MS/MS (R) for samples grown in AA, Glc, or Glc + AA. The plots from top to bottom show metabolite analysis for (A) AA vs. Glc (153 significantly up, 98 significantly down), (B) Glc + AA vs. Glc (192 significantly up, 106 significantly down), and (C) AA vs. Glc + AA (34 significantly up, 45 significantly down). Colors represent $\log_2$ fold-change (FC) with the darkest red and blue symbolizing the highest and lowest fold-change values, respectively. The top five metabolites were listed according to their $\log_2FC$ in both positive and negative directions. Letters G, H, and R in front of the metabolite name represent the acquisition method for the metabolite data. Metabolites with a $\log_2FC$ greater than 1 and FDR $P$-value ≤ 0.05 were considered significant. $n$ = 5 biological replicates for each condition, except for AA, where $n$ = 4. *National Center for Advancing Translational Sciences Chemical Genomics Center (NCGC) unique identifier for compound NCGC00385425-01_C21H36O11_.

condition but not in the AA-only condition. Lastly, glutamate displayed inconsistent consumption patterns across cultures.

## Differentially expressed genes reveal highly regulated pathways in *Histoplasma*

To explore how metabolite differences in *Histoplasma* grown on AA versus Glc arise, the transcriptome of *Histoplasma* incubated in the same media and growth conditions was analyzed. Out of 9,441 expressed genes, 326 were upregulated, and 779 were downregulated by at least four-fold during growth on AA compared to Glc (Fig. 3A). Of the differentially expressed genes, 320 were confidently annotated through identification of homologous proteins via reciprocal top-hit via BLASTx search (Table S3). Among these annotated genes, 169 were upregulated, and 151 were downregulated (Fig. 3B). Further analysis of the top 10 upregulated and downregulated genes revealed that *Histoplasma* modulates genes associated with redox balance, sulfate assimilation, and aromatic AA catabolism in response to carbon sources. For redox balance, genes encoding pyruvate decarboxylase, alcohol dehydrogenase, and catalase isozyme *P* showed significant upregulation in yeasts grown in AA (62-, 37-, and 30-fold, respectively). Expectedly, in AA-containing media that lack sulfate, genes involved in inorganic sulfate assimilation were downregulated, including sulfate permease, choline sulfatase, O-acetylhomoserine sulfhydrylase, and the sulfite reductase alpha subunit (110-, 81-, 40-, and 36-fold, respectively). In aromatic AA catabolism, phenylacetate 2-hydroxylase and 4-hydroxyphenylpyruvate dioxygenase were upregulated 73- and 27-fold, respectively, and, notably, carbonic anhydrase was significantly downregulated 313-fold (Table S3).

## Intramacrophage growth comprises a unique metabolic fingerprint

Using the *in vitro* metabolome models to establish the targeted metabolomics, we identified metabolites of yeast within the macrophage phagosome. In total, 101 yeast metabolites were successfully quantified: 26 AA, 12 sugars and sugar alcohols, 43 organic acids and phosphorylated compounds, and 20 of the most differentially accumulated metabolites from the untargeted metabolomic study (Table S4). Unsurprisingly, PCA showed a unique central metabolome for intracellular yeasts dissimilar from the *in vitro* growth conditions (Fig. 4). PC 1 and 2 explained 48.7 and 22.4% of the variance, respectively, where PC 1 distinguished between intramacrophage proliferation and *in vitro* cultivation conditions, while PC 2 nearly separated each condition individually. *Histoplasma* grown on Glc + AA or solely AA had overlapping confidence intervals; thus, their central carbon metabolomes were similar.

An analysis of total metabolite levels across all classes (AA, organic acids/phosphorylated compounds, and sugars) revealed significant deficiencies in intracellular yeasts. The total moles of AA in phagocytosed yeasts constituted only 31.6–38.6% of the *in vitro* conditions (Table S4). Similar deficiencies were observed in phagocytosed *Histoplasma* for total organic acids and phosphorylated compounds. This depletion pattern was more pronounced for total sugars, with intracellular yeast sugar levels reaching only 11.6 and 12.5% of those cultivated in Glc and Glc + AA, respectively (Table S4). These deprivations were particularly evident in the top two most abundant metabolites that were shared in all conditions: mannitol and glutamate. Mannitol levels were measured at 662.5, 645.8, 354.8, and 63.7 nmol mg DW$^{-1}$ under Glc, Glc + AA, AA, and intracellular growth, respectively, and glutamate levels were found to be 156.1, 142.5, 127.9, and 47.4 nmol mg DW$^{-1}$ within the respective conditions (Table S4). This suggests the phagosome environment may be more nutrient-limiting in comparison to nutrient-replete *in vitro* growth media.

To provide more comprehensive insights beyond individual metabolite fluctuations, metabolic pathway analysis was employed to visualize analyte levels within a unified metabolic map (Fig. 5). This approach contextualized the observed metabolite differences in relation to their pathways. While many metabolites showed significantly

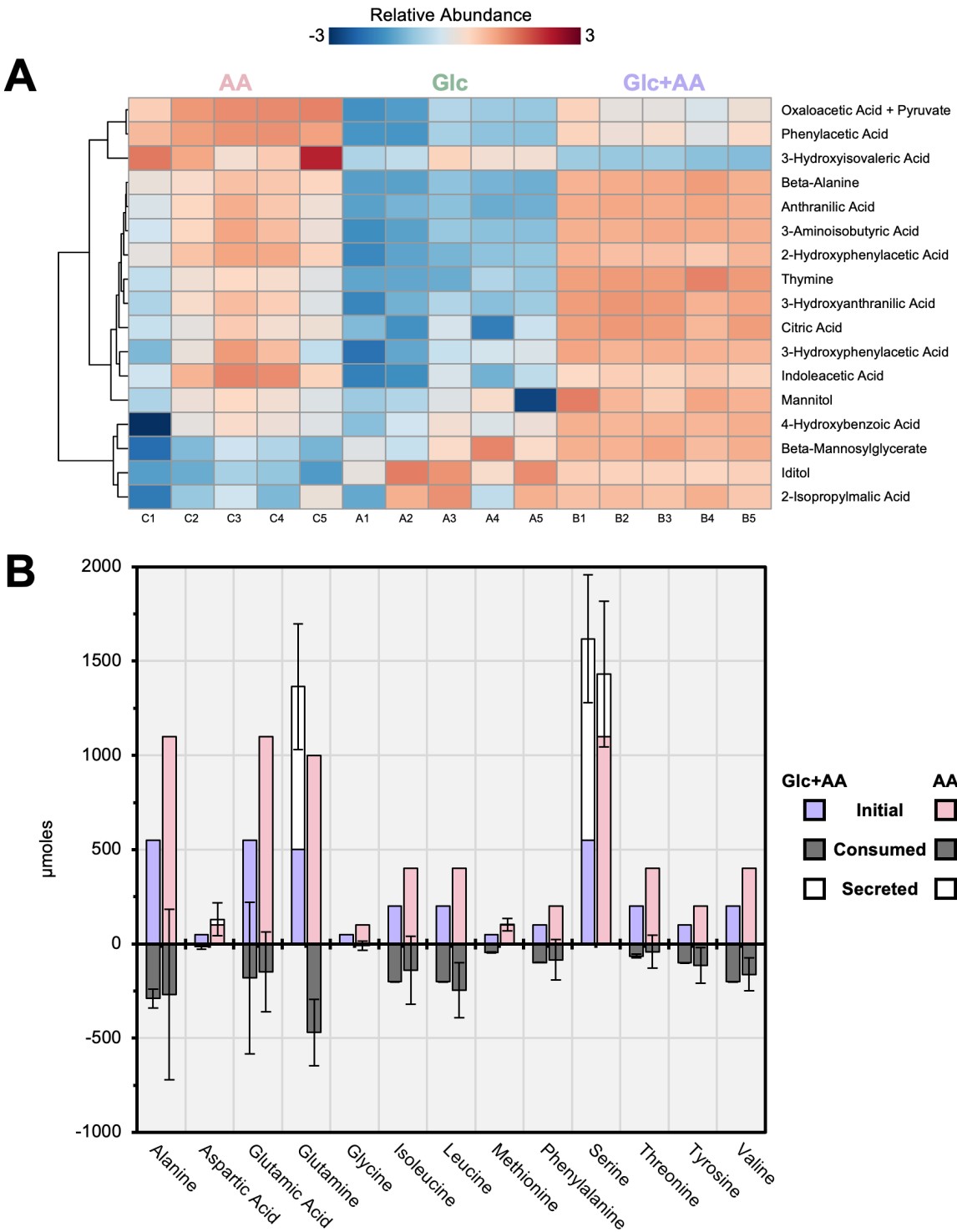

**FIG 2** *Histoplasma* secretes amino acid and sugar derivatives. (A) Hierarchical clustering heatmap with analysis of variance represents normalized intensity of significantly secreted metabolites that were at least 10-fold higher in intensity compared to uninoculated media for the respective growth condition (Glc, glucose; AA, amino acids; Glc + AA, glucose and amino acids). (B) Consumption or secretion of detectable AA was measured from initial quantities for the respective AA-containing media. Colored, dark, and white fill represent initial amount, consumption, and secretion of AA, respectively. Error bars represent SD of consumption or secretion and are of *n* = 5 biological replicates for each condition.

lower quantities in phagocytosed yeasts, three stood out with substantial increases compared to all *in vitro* conditions: cysteine/cystine, glucosamine 6-phosphate, and

phenylalanine, which were higher by at least 18.1-, 2.4-, and 2.3-fold, respectively. Interestingly, cystathionine, the biosynthetic precursor to cysteine, was enriched in AA-grown yeasts but not in phagocytosed yeasts. Glucosamine 6-phosphate levels were especially elevated in phagocytosed yeasts, with intracellular levels reaching 9.2 and 13.6 times those observed specifically in Glc or Glc + AA-supplemented conditions, respectively. A similar pattern was found for glucosamine 1-phosphate, though not for other directly related metabolites. Phenylalanine was the only aromatic AA present at high levels in phagocytosed yeasts despite its product, tyrosine, not being similarly abundant. In contrast, the three most significantly depleted metabolites in phagocytosed yeasts compared to *in vitro* conditions were homoserine, gamma-aminobutyric acid, and 2-isopropylmalate, with reductions of 142.4-, 98.0-, and 186.0-fold, respectively. Although no clear pattern emerged for the drastic deficiency of homoserine, gamma-aminobutyric acid levels were notably low, and its precursor, glutamate, was also significantly decreased. Additionally, 2-isopropylmalate, abundant under Glc-based growth, was highly reduced in all other growth conditions, even as leucine levels remained similar across growth conditions. Together, these findings highlight the significant metabolic adaptations that *Histoplasma* undergoes during intracellular growth, suggesting key compounds that may support yeast proliferation within the nutrient-limited environment of the macrophage phagosome.

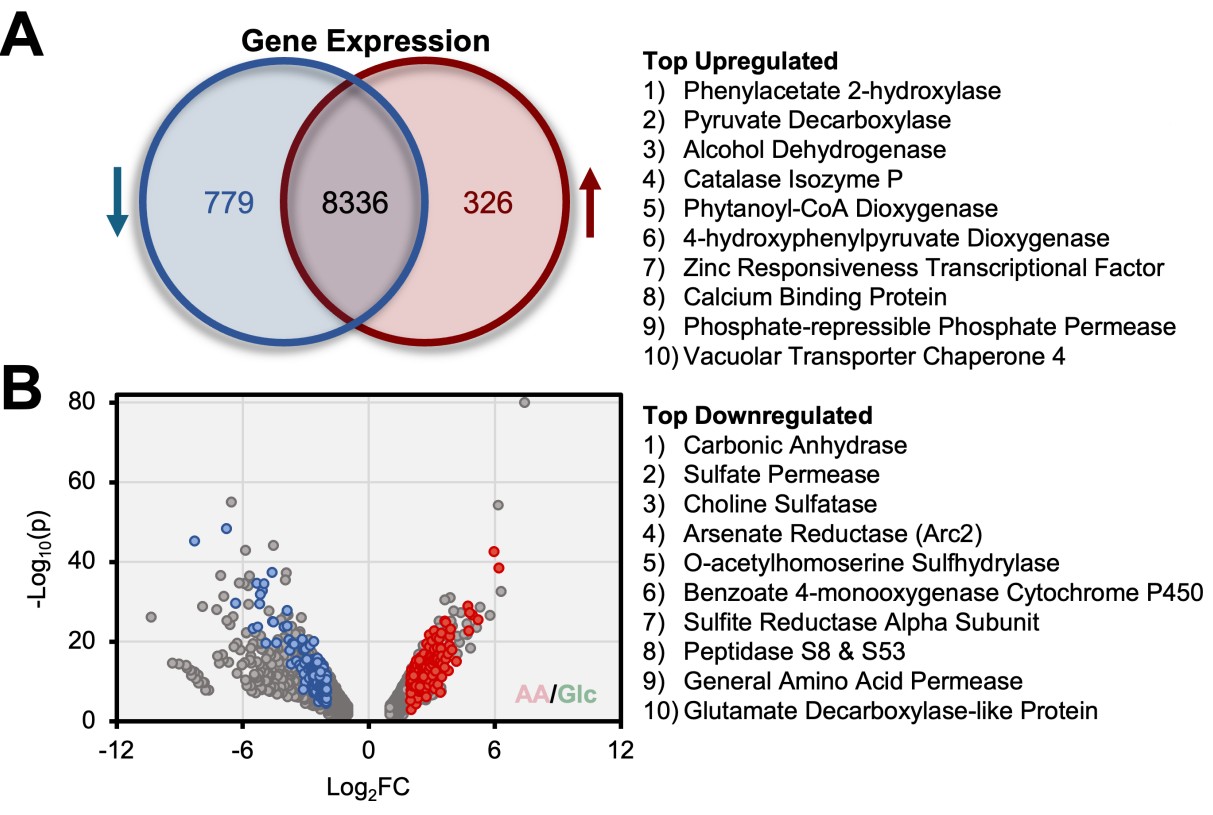

**FIG 3** *Histoplasma* regulates genes involved in redox balance, sulfate assimilation, and aromatic amino acid catabolism in response to growth on amino acids versus glucose. (A) Total expressed genes that were upregulated or downregulated by a $\log_2$ fold-change (FC) of 2, with amino acid cultivation compared to glucose cultivation. Blue, red, and black text represent the number of downregulated genes, upregulated genes, and no differential expression, respectively. (B) Volcano plot of expressed genes with a $\log_2$FC greater than 1 or less than −1. Gray dots indicate genes that were unannotated by the NCBI database. Colored dots designate annotated genes with a $\log_2$FC greater than two or less than −2. The top 10 upregulated and downregulated genes according to $\log_2$FC were listed.

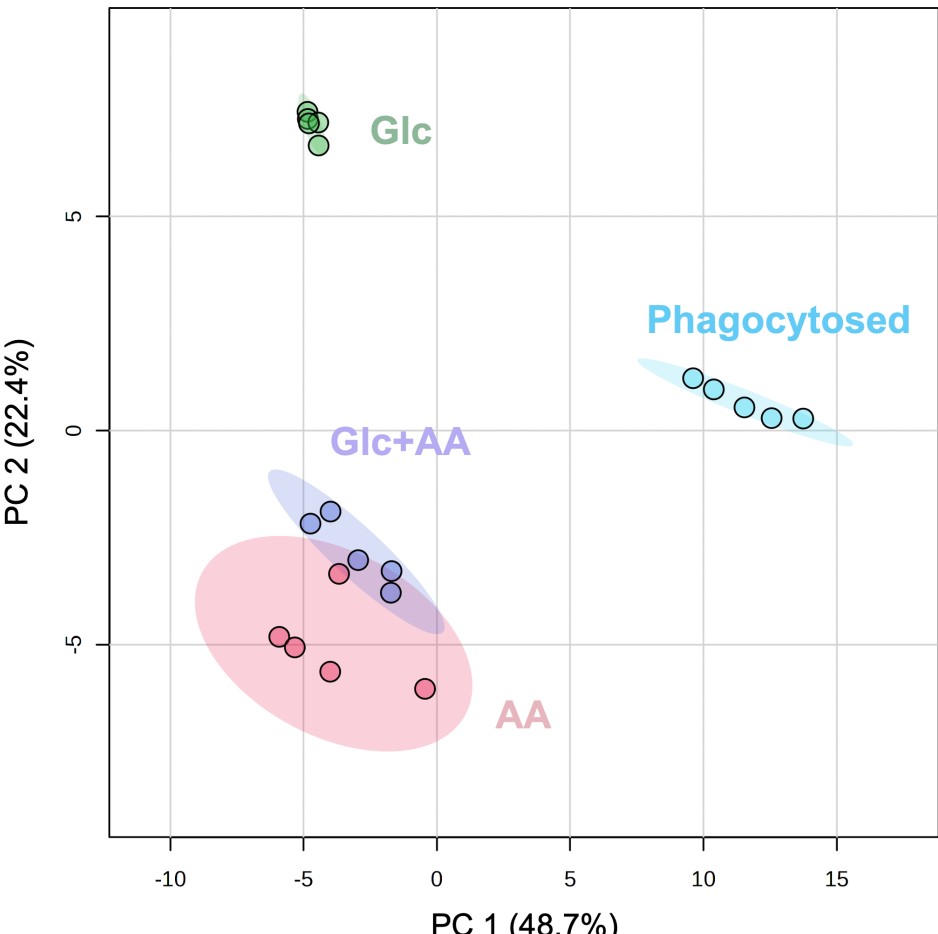

**FIG 4** The metabolome of phagocytosed *Histoplasma* does not resemble *in vitro* growth. Principal component analysis of targeted metabolomic data for *Histoplasma* cultured in different carbon sources (Glc, glucose; AA, amino acids; Glc + AA, glucose and amino acids) or extracted from macrophages after 24 h (phagocytosed). Shaded regions in the principal component analysis plot represent 95% confidence intervals. $n = 5$ biological replicates for each condition.

## DISCUSSION

In this study, the intracellular and extracellular metabolomes of *Histoplasma* grown *in vitro* on different carbon sources were explored using untargeted metabolomics to identify differentially accumulated metabolites. These metabolites, along with a library of other compounds, were then targeted for quantification in intracellular yeasts to infer characteristics of the macrophage phagosome.

### Limitations

While the untargeted metabolomics approach employed in this study provided valuable insights into the intracellular and extracellular metabolomes of *Histoplasma*, certain limitations must be acknowledged. A primary limitation is that while metabolic products have been identified, their biological relevance remains uncertain. The descriptive nature of this study necessitates further investigations to determine the functional significance of the detected metabolomes. Another challenge stems from the inability to confidently annotate every detected feature due to the lack of comprehensive spectral databases and variations in chromatography and mass spectrometry acquisition methods. Even

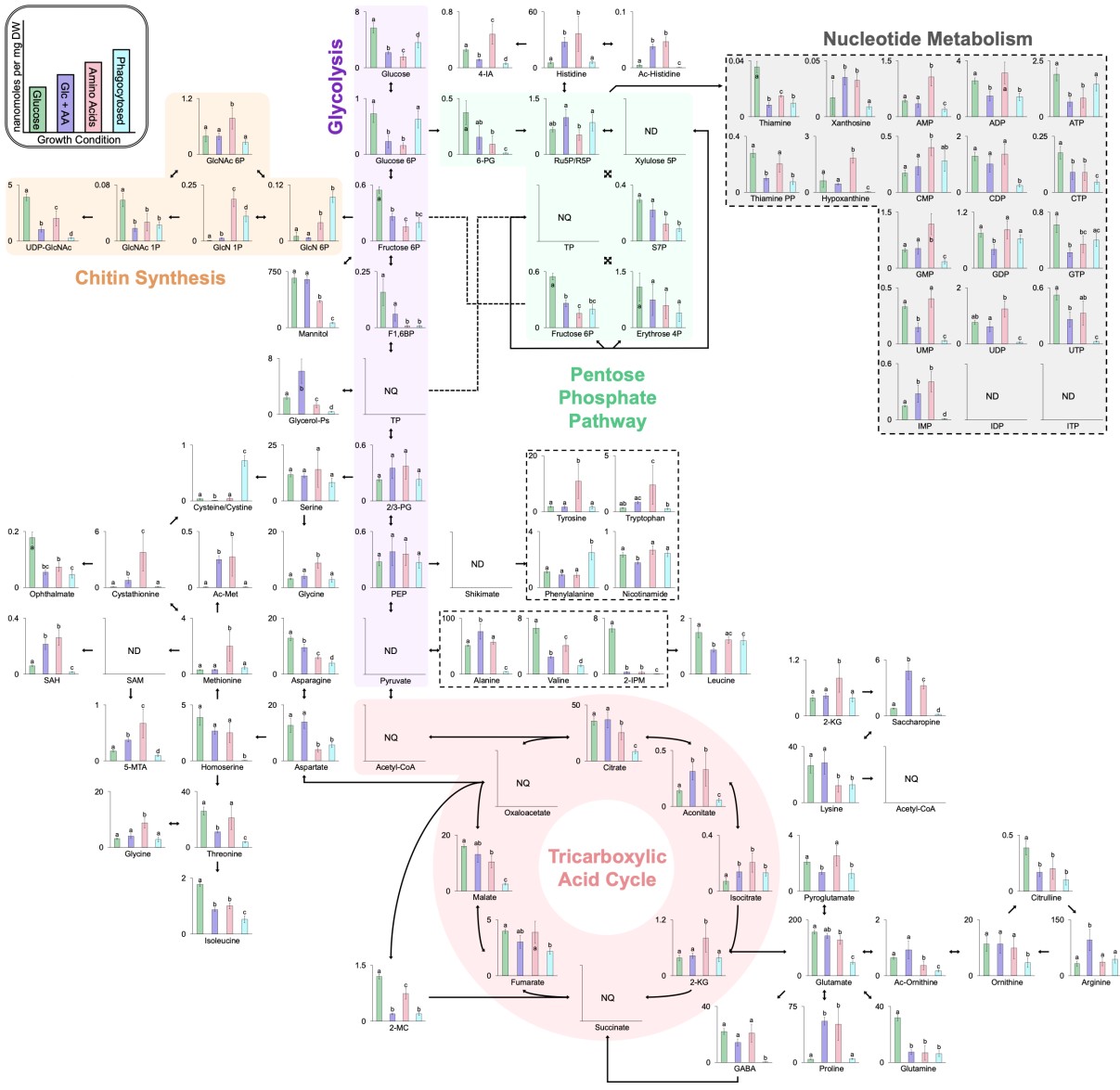

**FIG 5** Intracellular *Histoplasma* exhibits metabolic deficiencies despite cysteine/cystine, leucine, and phenylalanine being available inside macrophages. The metabolic map represents metabolite quantities of *Histoplasma* cultured in different growth conditions or extracted from macrophages after 24 h. Values are expressed in nanomoles per milligram dry weight of *Histoplasma*. Unidirectional arrows represent enzymatic reactions that are irreversible or have no equivalent enzyme that catalyzes the reverse reaction. Metabolites represented as NQ were not quantifiable due to poor ionization efficiency, spontaneous degradation, or contamination. Metabolites represented as ND were found to be below the limit of quantification. Error bars represent SD and are of $n = 5$ biological replicates for each condition. Bars with different letters are significantly different according to parametric analysis of variance post-hoc analysis using Tukey's HSD with raw $P \leq 0.05$. 2/3 PG, 2 & 3-phosphoglycerate; 2-IPM, 2-isopropylmalate; 2 KG, 2-ketoglutarate; 2-MC, 2-methylcitrate; 4-IA, 4-imidazoleacetate; 5-MTA, 5-methylthioadenosine; 6 PG, 6-phosphogluconate; Ac-Histidine, acetyl-histidine; Ac-Ornithine, acetyl-ornithine; ADP, adenosine diphosphate; AMP, adenosine monophosphate; ATP, adenosine triphosphate; CDP, cytidine diphosphate; CMP, cytidine monophosphate; CTP, cytidine triphosphate; Erythrose 4P, erythrose 4-phosphate; F1,6BP, fructose 1,6-bisphosphate; Fructose 6P, fructose 6-phosphate; GABA, gamma-aminobutyric acid; GDP, guanosine diphosphate; GlcN 1P, glucosamine 1-phosphate; GlcN 6P, glucosamine 6-phosphate; GlcNAc 1P, N-acetylglucosamine 1-phosphate; GlcNAc 6P, N-acetylglucosamine 6-phosphate; Glucose 6P, glucose 6-phosphate; Glycerol-Ps, glycerol phosphates; GMP, guanosine monophosphate; GTP, guanosine triphosphate; IDP, inosine diphosphate; IMP, inosine monophosphate; ITP, inosine triphosphate; PEP, phosphoenolpyruvate; Ru5P/R5P, ribulose 5-phosphate/ribose 5-phosphate; S7P, sedoheptulose 7-phosphate; SAH, S-adenosylhomocysteine; SAM, S-adenosylmethionine; Thiamine PP, thiamine pyrophosphate; TP, triose phosphate; UDP, uridine diphosphate; UDP-GlcNAc, uridine diphosphate N-acetylglucosamine; UMP, uridine monophosphate; UTP, uridine triphosphate; Xylulose 5P, xylulose 5-phosphate.

with high spectral matching to a reference spectrum, an annotation does not guarantee the identification is correct, as isomers, particularly among sugars, are likely to ionize and

fragment in the same manner. In this study, for instance, while it is possible for flavonoids to be produced by yeasts (26), sakuranetin represents an example of potential but not definitive identification. Variability in ionization among molecules further complicates analyses, for example, affecting the significance of certain metabolites and making it difficult to distinguish between background noise and meaningful secretion intensities. Additionally, the use of Glc media supplemented with ammonium sulfate as the nitrogen source may have introduced excess nitrogen, potentially influencing metabolic profiles and confounding interpretations, particularly with respect to nitrogen-sensitive pathways that intersect with central carbon metabolism. Many of these challenges underscore the necessity for the integration of complementary approaches, such as targeted metabolomics, to reduce some of these limitations, although the use of targeted metabolomics may narrow the scope of analyses to a smaller portion of the metabolome.

## *Histoplasma* engages in cellular respiration rather than fermentation

As reported in this study, *Histoplasma*'s metabolome and transcriptome undergo substantial shifts in response to different milieus, including the intracellular phagosomal environment. A significant factor driving these metabolic changes may be the yeast's primary metabolic mode. This raises questions about whether glycolytic products are predominantly fermented, even in the presence of oxygen (Crabtree-positive), or enter the tricarboxylic acid cycle for cellular respiration (Crabtree-negative). Evidence suggests that *Histoplasma* behaves as a Crabtree-negative yeast. For example, Glc-based growth downregulates enzymes involved in alcoholic fermentation (Fig. 3), indicating the absence of the Crabtree effect. Ethanol was also undetectable in the medium after 68 h of *in vitro* growth (data not shown), regardless of carbon source, demonstrating that cellular respiration is the primary energy-generating pathway *in vitro*. Furthermore, genetic and transcriptomic studies indicate that intracellular growth of *Histoplasma* relies upon metabolism of gluconeogenic substrates obtained from the phagosome. However, genes encoding glyoxylate cycle enzymes, a pathway important for incorporation of C2 compounds, were not upregulated during infection, suggesting fatty acids do not supply necessary carbon (22). Together, this suggests that *Histoplasma* likely engages in cellular respiration of AA during intracellular growth. While these observations are primarily contextualized on *in vitro* data, they provide valuable foundational knowledge on *Histoplasma*'s metabolic mode. Understanding the yeast's reliance on cellular respiration may inform future research on pathogen survival mechanisms within host cells.

## Metabolites contributing to *Histoplasma's* pathogenicity

Numerous metabolites were found to be significantly excreted by *Histoplasma* yeasts, some of which could contribute to *Histoplasma*'s pathogenicity. First, mannitol emerged as a significantly secreted metabolite (Fig. 2A), consistent with its presence in extracellular vesicles released from *Histoplasma* (27). Investigations into mannitol's cellular functions in other fungi have demonstrated that mutants lacking mannitol biosynthesis or catabolism exhibit reduced virulence in both plant and animal tissues (28–33), with evidence for mannitol mitigating oxygen radicals (34–36). *Histoplasma* yeast production of mannitol may provide a non-enzymatic method of reactive oxygen species detoxification in combination with production of redundant catalases (CatB and CatP) (16). Alternatively, mannitol may provide for excess carbon storage, although this may only apply to yeast grown in plentiful nutritional substrates *in vitro* since mannitol levels were exceptionally low in yeasts within the macrophage phagosome. In this light, metabolites observed to be "low" in yeasts within macrophages may result from exceptionally high levels produced by yeast during *in vitro* culture in excess carbon.

The second finding revealed through extracellular untargeted metabolomics identified a notable secretion of aromatic AA derivatives by *Histoplasma* (Fig. 2A). Among these secreted compounds, 3-hydroxyanthranilate and anthranilate, both derived from tryptophan, could play nonenzymatic roles in reducing ferric ($Fe^{3+}$) to ferrous ($Fe^{2+}$)

iron, thereby facilitating iron acquisition critical for biological processes (37–40). This mechanism aligns with previous studies demonstrating ferric iron-reducing activity in low-molecular weight culture supernatants (15) and anthranilate discovery within extracellular vesicles released from *Histoplasma* (27). The presence of anthranilates may also explain why siderophore-deficient strains persist in mouse lungs post-infection (14). It is plausible, though, that a lack of available iron is not immediately lethal, allowing yeasts to persist in an unreplicating state. For other exported derivatives, such as phenylacetate and its hydroxy-relatives, secretion may represent an avenue to remove toxic metabolic byproducts, as phenylacetate is a known antimicrobial (41–43). Significantly increased intracellular phenylalanine abundance (Fig. 5) and upregulated aromatic AA catabolism genes (Fig. 3) suggest intracellular yeasts catabolize aromatic AA and/or secrete these antimicrobials to counteract their toxicity, a strategy described previously (44). Overall, the significant secretion of anthranilates and phenylacetates potentially indicates a concerted strategy to assimilate aromatic AA and convert them into derivatives that could support iron acquisition in the case of tryptophan.

## Carbon assimilation by *Histoplasma* inside macrophage phagosomes

To proliferate during macrophage infection, *Histoplasma* must acquire carbon and other necessary elements from sources available inside macrophage phagosomes. Previous work has established phenylalanine and tyrosine as available to *Histoplasma* within macrophage phagosomes (25). In this study, intracellular yeasts showed a significant increase in phenylalanine abundance, whereas tyrosine levels were not (Fig. 5). This discrepancy may be due to a smaller available pool of tyrosine, as total consumption of phenylalanine and tyrosine during *in vitro* growth did not differ significantly, arguing the differences do not stem from differences in uptake capability (Fig. 2B).

In contrast, analysis of leucine levels across all conditions revealed no significant differences (Fig. 5). However, 2-isopropylmalate, a precursor in leucine biosynthesis, was abundant only during Glc cultivation, when leucine biosynthesis is required, but its low abundance during intracellular growth suggests that *Histoplasma* obtains leucine from the phagosomal space instead of synthesizing it *de novo*. In support of this, yeasts readily take up leucine when grown *in vitro* in media containing leucine (Fig. 2B). These findings suggest that intracellular yeasts may utilize leucine in the phagosome as a source of carbon and nitrogen for proliferation.

In addition to leucine, *Histoplasma* likely assimilates cysteine/cystine from the phagosomal environment. The present study showed significantly higher cysteine/cystine levels within intracellular yeasts than in any other condition (Fig. 5). Since *Histoplasma* yeasts are auxotrophic for cysteine (18–21), the presence of abundant cysteine/cystine intracellularly indicates uptake rather than biosynthesis. It is worth noting that during boiling water extraction, cysteine is oxidized into cystine (45); thus, no cysteine was actually detected in any condition, but the elevated cystine levels likely reflect the presence of abundant cysteine within intracellular yeasts. Such high levels likely also explain why homoserine is completely abolished with intracellular growth (Fig. 5), as cysteine competitively inhibits homoserine dehydrogenase (46, 47), the only enzymatic reaction producing homoserine. These data, together with the finding that cysteine but not its precursor, cystathionine, is high in intracellular yeasts (Fig. 5), support cysteine/cystine as a nutrient and sulfur source consumed by *Histoplasma* in the phagosomal environment.

In conclusion, this study provides a detailed exploration of *Histoplasma*'s metabolome, uncovering key metabolites and pathways linked to *Histoplasma*'s survival and growth within murine macrophages. Metabolites identified could serve dual functions: nutritional metabolism, as well as production of compounds to facilitate *Histoplasma*'s intracellular lifestyle through reactive oxygen species management and iron acquisition. Future research investigating the functional significance of identified metabolites through loss of biosynthesis or metabolism of these molecules will be instrumental in assessing their impact on and roles in macrophage infection. This and further

metabolomic studies underscore the importance of metabolic adaptations to the phagosomal environment and highlight the increasing potential of nutritional immunity avenues that could potentially be exploited for novel therapeutic approaches for histoplasmosis and related intracellular infections.

## MATERIALS AND METHODS

### Chemicals

[U-$^{13}$C]Glycine, [U-$^{13}$C]fumarate, and [U-$^{13}$C]benzoic acid were obtained from Cambridge Isotope Laboratories (Tewksbury, MA). Methoxyamine hydrochloride and pyridine were purchased from Sigma-Aldrich (St. Louis, MO). Methylene chloride, chloroform, N-methyl-N-trimethylsilyltrifluoroacetamide plus 1% trimethylchlorosilane (MSTFA + 1% TMCS), and LC–MS-grade solvents, such as acetonitrile, methanol, water, ammonium formate, and formic acid, were purchased from Thermo Fisher Scientific (Waltham, MA). Metabolite standards were ordered from Sigma-Aldrich, Thermo Fisher Scientific, Cayman Chemical, Toronto Research Chemicals, BOC Sciences, and Benchchem.

### *Histoplasma capsulatum* strains and cultivation

This study utilized the *Histoplasma capsulatum* wild-type G217B clinical isolate (NAm2 clade). *Histoplasma* was cultured as yeasts at 37°C using *Histoplasma* macrophage medium (HMM) for pregrowth cultures, and washed yeasts were transferred to a defined 3M-based medium supplemented with carbon and nitrogen sources as described below (48). Yeast liquid cultures were grown at 37°C with continuous shaking (200 rpm) and reached late exponential growth phase (68 h) before collection. *Histoplasma* was plated on HMM media solidified with 0.6% agarose and supplemented with 25 µM FeSO$_4$ for general maintenance. Uninoculated cultures were incubated alongside inoculated cultures to control for evaporation of liquid.

### Growth of *Histoplasma* yeasts for metabolite analysis

For Glc cultivation, wild-type *Histoplasma* yeasts were cultured in 3M media containing 50 mM Glc and 15 mM ammonium sulfate. For AA cultivation, yeasts were cultivated in 3M minimal media with an AA cocktail as the carbon and nitrogen sources (10 mM each of glutamate, glutamine, alanine, serine, and proline). This medium also included Gibco minimum essential medium (MEM) amino acid solution and Gibco MEM non-essential amino acid solution, both at 10× the recommended concentration, and adjusted to pH 6.5. Final concentrations were at 11 mM alanine, 6 mM arginine, 1 mM asparagine, 1 mM aspartate, 1 mM cystine, 11 mM glutamate, 10 mM glutamine, 1 mM glycine, 2 mM histidine, 4 mM isoleucine, 4 mM leucine, 4 mM lysine, 1 mM methionine, 2 mM phenylalanine, 11 mM proline, 11 mM serine, 4 mM threonine, 0.5 mM tryptophan, 2 mM tyrosine, and 4 mM valine. The Glc + AA growth condition was a 1:1 mixture of both media types used. For all treatments, yeasts were cultivated for 68 h at 37°C with shaking at 200 rpm. AA and Glc + AA cultures were grown at a volume of 12.5 mL. Due to slower growth in the Glc condition (Fig. S3A), Glc cultures were grown at a volume of 50 mL to collect the required number of yeasts for metabolomic analyses. Despite differences in growth rates, analysis of various housekeeping genes demonstrated consistent expression across conditions (Fig. S3B).

Following cultivation, yeast cultures and supernatant were separated by centrifugation (5 min at 5,000 × *g*). Culture supernatants were filtered and frozen in liquid nitrogen and stored at −80°C for extracellular untargeted metabolomic analysis. Uninoculated media incubated alongside inoculated cultures were also collected and flash-frozen before lyophilization. The yeast cell pellets were washed twice with cold sterile water, resuspended in 0.5 mL of 60% aqueous methanol to devitalize *Histoplasma*, rapidly frozen in liquid nitrogen, and stored at −80°C.

## Isolation of intracellular yeasts

A P388D1-derived murine macrophage cell line (49) was utilized for isolation of intracellular yeasts. Macrophages were cultivated at 37°C with 5% $CO_2$/95% air in Ham's F-12 media (Corning) containing 10% fetal bovine serum (FBS, Sigma). Next, $1 \times 10^7$ macrophages were infected with *Histoplasma* yeasts at a multiplicity of infection (yeasts:macrophages) of 2:1. After incubation for 2 h, non-internalized yeasts were removed by replacement of the medium with fresh Ham's F-12 with 10% FBS. Intracellular yeasts were isolated 24 h post-infection by washing macrophages twice with 5 mL phosphate-buffered saline, followed by lysis of macrophages with 10 mL cold water. Cells were harvested from the wells using a cell scraper, and the lysate passed through a 25-gauge needle four times to ensure macrophage lysis and release of intracellular yeasts. Yeasts were collected by centrifugation at 4°C (5 min at 3,000 × *g*), washed twice with 5 mL ice-cold sterile water, and resuspended in 0.5 mL of 60% aqueous methanol to devitalize *Histoplasma*. Samples were subsequently frozen in liquid nitrogen and stored at −80°C.

## Intracellular metabolite extractions for untargeted metabolomics analyses

Dual sets of *Histoplasma* samples for each growth condition (five biological replicates each) underwent separate extractions for determination of the *Histoplasma* metabolome. A modified CMW phase extraction was performed, as previously detailed (50), to extract polar and semi-polar compounds from *Histoplasma*. Before disruption by bead-beating, [U-$^{13}$C]glycine (100 nmol) was added as an internal standard. After disruption, a "partial" CMW solution was added to the methanol and water already present to obtain a final CMW mixture of 1:2.5:1 (v/v/v). Phase separation was achieved by adding 400 µL of cold ultrapure water, and 500 µL of the upper polar phase was transferred to a 1.5 mL microcentrifuge tube. Subsequently, 500 µL of the upper phase was progressively transferred to a 250 µL deactivated glass insert via successive rounds of speed-vacuuming at 30°C until the final volume was about 200 µL of water. Extracts were flash-frozen in liquid nitrogen and lyophilized at −80°C for 3–4 h.

A BW extraction, following the method described (51, 52), was performed. [U-$^{13}$C]Glycine (100 nmol) was supplemented to all samples, except blanks. Filtered extracts were split for LC–MS/MS and GC–MS analyses, flash-frozen, and lyophilized. Following lyophilization, samples for GC–MS were resuspended with 400 µL cold 50% aqueous methanol. Then, extracts were progressively transferred to 250 µL deactivated glass inserts after undergoing consecutive rounds of speed-vacuuming at 30°C. Samples were flash-frozen in liquid nitrogen and lyophilized at −80°C for 3–4 h.

## GC–MS analysis

CMW and BW extracts and *Histoplasma* growth media underwent methoximation–silylation derivatization (50). For both uninoculated and 68 h-inoculated media, 10 µL of sample was added to a 250 µL glass insert and lyophilized for 3–4 h prior to methoxymation and derivatization. Derivatized extracts and media were analyzed using a Thermo Trace 1310 gas chromatograph coupled to an ISQ single quadruple mass spectrometer (Thermo Fisher, San Jose, CA, USA). QCs were prepared by transferring 5 µL of each derivatized biological sample to a 250 µL glass insert. The initial oven temperature was set at 70°C for 5 min. For intracellular metabolite analysis, the following oven temperature ramps were applied: 70–235°C at 3°C min$^{-1}$ and held for 5 min; 235–320°C at 6°C min$^{-1}$ and held for 5 min; and 320–330°C at 10°C min$^{-1}$ and held for 5 min. For media analysis, the following oven temperature ramps were applied: 70–220°C at 3°C min$^{-1}$ and 220–320°C at 65°C min$^{-1}$ and held for 3.5 min. Ion source temperature for both intracellular and extracellular metabolite analyses was set at 300°C, and MS spectra were acquired over m/z 50–1100 amu with a scan time of 81 ms. A split ratio of 7.5 was selected for the injection mode. The QCs were injected multiple times throughout the run to make sure that the equipment was functioning correctly. A 50 µg/mL alkane

standard mixture from Sigma-Aldrich (St. Louis, MO), spanning seven to 40 carbons in length, was also injected for use in establishing the Kováts retention index.

## GC–MS headspace sample preparation and analysis

To assess the production and secretion of ethanol and acetate, headspace GC–MS analysis was conducted. 2 g of sodium chloride was placed into each 20 mL GC–MS headspace vials (Thermo Fisher Scientific, Waltham, MA) to induce volatility. Then, 4.990 mL of inoculated media and 10 µL of a 0.2% (v/v) aqueous acetonitrile solution (internal standard) were added to the 20 mL headspace vial, which was promptly capped. A blank was prepared by equally mixing in volume the three different unino-culated growth media and adding 5 mL to a vial. An external standard mix of 0.07% (v/v) acetonitrile, 0.07% (v/v) ethanol, and 33.33% (v/v) acetic acid was prepared. Then, 15 µL of the external standard mix was added to a 4.985 mL mixture of uninoculated media within a vial. All vials were vigorously vortexed for 1 min and placed in the GC–MS autosampler apparatus.

Volatile organic compounds (VOCs) were detected and quantified using a Trace 1310 gas chromatograph system coupled to an ISQ single quadrupole mass spectrometer (Thermo Fisher Scientific, Waltham, MA). 20 mL vials containing 5 mL of medium spiked with acetonitrile were incubated at 80°C for 10 min with 10 s shaking/10 s pause cycle using a Triplus RSH autosampler. 500 µL of extracted VOC sample was injected to a split/splitless injector (225°C). A split ratio of 5:1 was applied to the extract. The VOCs were resolved using a DB-624 UI (30 m × 0.25 mm × 1.4 µm) column under a constant flow of helium setup at 1.6 mL/min. The GC conditions were as follows: the initial temperature ramp was set to 32°C with a 2 min hold; the first temperature ramp was 3°C/min up to 40 °C with no hold; the second temperature ramp was 50°C/min up to 230°C with 1 min hold; and the third temperature ramp was 50°C/min up to 250°C with a 5 min hold. The total GC–MS run was 15 min.

For MS analysis, VOCs were ionized using electron impact (EI) ionization in positive ion mode (200°C transfer line and 230°C EI source). A mixture of external standards (acetate, acetonitrile, ethanol) was utilized to record the retention time of each ionized metabolite that was scanned over a mass range of 20–200 amu. Once the compounds had assigned retention time, the most abundant product ion for each analyte was selected to perform single ion monitoring. In this study, product ions for acetate, acetonitrile, and ethanol were 60, 41, and 45 amu, respectively. The VOCs were acquired using Xcalibur 2.2 (Thermo Fisher Scientific, Waltham, MA) version, and the data were processed through FreeStyle 1.7 (Thermo Fisher Scientific, Waltham, MA) version.

## Resuspension for high-resolution LC–MS/MS analysis

For both CMW and BW samples intended for high-resolution (HR) LC–MS/MS analysis, dried extracts were resuspended with 400 µL of 50% aqueous methanol. Subsequently, the sample volume was split equally into 2 mL microcentrifuge tubes, flash-frozen in liquid nitrogen, and lyophilized again. Samples were resuspended with 100 µL of an internal standard mixture, except blanks, to account for retention time deviations during data analysis. The internal standard mixture for injection onto the RP column consis-ted of 3-(morpholin-4-yl)propane-1-sulfonic acid (50 nmol), 5-fluorocytosine (250 nmol), [U-$^{13}$C]fumarate (1250 nmol), trans-zeatin (25 nmol), ampicillin (50 nmol), [U-$^{13}$C]benzoic acid (250 nmol), trans-cinnamic acid-d6 (250 nmol), and 9-phenanthrol (50 nmol) in a 20% aqueous methanol solution. For injection onto the HILIC column, the same internal standards and their respective amounts were resuspended in an 80% aqueous acetoni-trile solution. QCs for each sample set were then made by adding 10 µL from each biological replicate. True blanks were resuspended with only the corresponding aqueous solution without internal standards. 3 and 4 µL of sample were injected onto the RP and HILIC columns, respectively.

## Q-TOF LC–MS/MS conditions for untargeted metabolomics

Resuspended extracts were analyzed using an Exion ultra high-performance liquid chromatography system coupled with a high-resolution mass spectrometer Triple-TOF6600+ from AB Sciex (Framingham, MA). RP and HILIC chromatographic separations (orthogonal separations) provided complementary information about the biological samples. The temperature of the autosampler was maintained at 10°C for both liquid chromatography techniques.

For RP chromatography, compounds were separated using a Kinetex F5 column (150 × 2.1 mm, 2.6 µm) with a security guard column F5 (10 × 2.1 mm) from Phenomenex (Torrance, CA). Column temperature was set up at 20°C, and metabolites were eluted at a flow rate of 0.2 mL/min. A gradient made of 0.25% (v/v) of formic acid plus 5 mM ammonium formate in water (Solvent A) and 0.1% (v/v) formic acid in acetonitrile (Solvent B) was applied as follows: 0–2.0 min, 0% B; 2.0–18.0 min, 0-80% B; 18.0–18.1 min, 80-95% B; 18.1–21.0 min, 95% B; 21.0–21.1 min, 95-0% B; and 21.1–25.0 min, 0% B.

For HILIC separation, compounds were resolved using an ACQUITY Premier BEH Amide VanGuard FIT (150 × 2.1 mm, 1.7 µm) column associated with an ACQUITY Premier BEH Amide pre-column (5 × 2.1 mm, 1.7 µm) from Waters (Milford, MA). For HILIC elution, a gradient of 0.2% (v/v) of formic acid plus 25 mM ammonium formate in water (Solvent A) and 0.15% (v/v) formic acid plus 10 mM ammonium formate in 90% aqueous acetonitrile (Solvent B) was employed, with a flow rate of 0.3 mL/min. The following gradient was applied: 0–2.00 min, 100% B; 2.00–6.00 min, 100-70% B; 6.00–9.35 min, 70-40% B; 9.35–11.00 min, 40-30% B; 11.00–13.50 min, 30-100% B; and 13.50–20.00 min, 100% B. The column temperature was maintained at 35°C.

## High-resolution mass spectrometry (triple TOF)

Data-independent acquisition using sequential window acquisition of all theoretical mass spectra (SWATH-MS) scan survey was utilized as previously described (53). Briefly, a data-dependent acquisition scan survey was performed on QCs (mixture of biological extracts) in the negative and positive modes for RP and HILIC separations to obtain precursor ion data, which were used to generate a total of 36 precursor ion variable SWATH-MS windows. Precursor ions were detected over a m/z range of 70–1250 amu in either negative or positive mode (RP and HILIC), with an accumulation time of 200 ms. MS/MS spectra were acquired over m/z 30–1250 amu with an accumulation time of 25 ms. The total cycling time was 1.15 s. The other mass spectrometer parameters related to the electrospray ionization source and the quadrupoles are reported in Table S5. A calibrant delivery system was used to deliver atmospheric-pressure chemical ionization negative or positive solution every eight samples to correct for any mass drift that may have happened during the LC–MS/MS run. The data were acquired using Analyst TF 1.8.1 software from AB Sciex (Framingham, MA).

## Untargeted metabolomic data processing

### GC–MS

For intracellular metabolites, GC–MS .raw files (Xcalibur v2.2) were converted to .abf files using the Reifycs Analysis Base File Converter and uploaded to MS-DIAL (v4.9221218) for data collection, peak detection, deconvolution, identification, alignment, and filtering (Table S6). A retention index file derived from the retention times of alkanes C10–C38 was created to assign retention indices to features for comparison, facilitating identification through the GCMS DB-Public-KovatsRI-VS3 spectral library. Features were manually curated to ensure proper alignment and resolution. Subsequently, normalization occurred by dividing all feature intensities by the intensity of the [U-$^{13}$C]glycine in the corresponding sample. Features with a signal/noise ratio less than 10 were excluded from analysis. In the final list of tentatively identified metabolites, any unknowns and duplicate annotations for the same metabolite were removed, where the duplicate feature with the highest intensity was kept. In extracellular untargeted metabolomic

analysis, the FC of normalized feature intensities in inoculated samples compared to uninoculated media was calculated to determine true secretion. An average FC value for a growth condition less than 10 was considered insignificant secretion and not treated as true secretion of the annotated metabolite.

### HR–LC–MS/MS

For untargeted metabolomic analysis, LC–MS/MS .wiff files (Analyst TF v1.8) were uploaded to MS-DIAL (v5.1.230429). Sequential window acquisition of all theoretical mass spectra was chosen as the acquisition type, and similar parameters were applied as in the GC–MS processing (Table S6). Retention time lengths and adduct selections were adjusted between data sets to accommodate RP and HILIC chromatography, as well as positive and negative ion modes. Spectral libraries used for identification consisted of an in-house library containing spectra of central metabolic intermediates, the Mass Bank of North America library, the Global Natural Products Social Molecular Networking library, and an *in silico* spectral library from the Mass Bank of North America. Chromatograms were aligned between samples with QCs, and retention time corrections for total ion chromatograms were made using the retention times from the internal standard mixture.

For curation of LC–MS/MS data, MS-CleanR was employed to remove features present in blanks at a 0.15 blank ratio, ghost peaks, and/or those with incorrect m/z values (54). Total scores were calculated based on retention time and spectra matching, and features with total scores less than 1.1 were removed. Features with a signal/noise less than 10 were also excluded. Unknowns and duplicates were removed following the same parameters as in GC–MS data curation. Positive and negative ion mode data sets were amalgamated using the same duplicate removal criteria. Metabolite names were standardized using MetaboAnalyst's 6.0 Pathway Analysis for uniformity across data sets, and feature intensities were normalized according to the intensity of [U-$^{13}$C]glycine in the corresponding sample. For PCA, all unknowns and duplicate features were retained to avoid bias in plotting the metabolomes (Fig. S2).

## Targeted metabolomic analyses

*In vitro*-grown (Glc, AA, Glc + AA) and phagocytosed *Histoplasma* samples were evaporated (speed-vac), flash-frozen, and lyophilized. Because BW extracts phosphorylated compounds much more efficiently (55), the BW extraction was carried out following the untargeted metabolomic methods (51). After filtering extracts, samples were again lyophilized and resuspended with 500 µL of cold water as previously described (50).

Intracellular metabolites were identified and quantified using an Agilent 1290 Infinity II HPLC coupled to an AB Sciex QTRAP6500 + mass spectrometer system as previously described (50, 51, 55–58). Standards for the significantly different metabolites from the untargeted metabolomic study were optimized via direct infusion as described (53) and incorporated into pre-existing targeted metabolomics LC–MS/MS methods. For sugars and sugar alcohols, two dilutions were prepared due to the oversaturation of mannitol: 1:500 in acetonitrile/water (70:30; v/v) and 1:10 in acetonitrile/water (70:30; v/v). For metabolites from phagocytosed yeasts, 1:5 dilutions were made in the acetonitrile solution. 1 µL of the 1:500 dilution, 10 µL of the 1:10 dilution, and 10 µL of the 1:5 dilution were injected onto the column. For analysis of AA, samples were diluted 1:50 in 1 mM hydrochloric acid (*in vitro*-grown yeast samples) or 1:5 (phagocytosed yeast samples). 3 µL of the 1:50 dilution and 10 µL of the 1:5 dilution were injected onto the column. For organic acids and phosphorylated compounds, samples were diluted 1:10 in water (*in vitro*-grown yeast samples) or left undiluted (phagocytosed yeast samples), and 5 µL was injected onto the column. Data were acquired and processed using Analyst v. 1.7 and MultiQuant v. 3.0 software, respectively. To quantify metabolites, each peak area was adjusted by dividing the area by the ratio of the $^{13}$C-glycine area in the samples to that of the internal standard alone. This corrected area was multiplied by the total sample dilution, which was compared to the concentration and peak area of the corresponding external standard to determine the quantity of metabolite in moles (50, 57, 58).

## Transcriptomic data processing

RNAseq data (accession #: PRJNA1069744) was compiled as expression changes ($\log_2$FC) comparing AA vs. Glc. Genes with a $\log_2$FC greater than two or less than −2 were used as queries for BLASTx searches and matches to *Saccharomyces* genes used to name the corresponding *Histoplasma* gene. The reciprocal top hit was used for gene annotation if the percent identity exceeded 90%. In cases where a query produced two distinct results with percent identities greater than 90%, the top two hits were listed (Table S3).

## Statistical analysis

Statistical analyses were performed using MetaboAnalyst 6.0. Prior to statistical analyses, data were not filtered using reliability, variance, and abundance filters but were log-transformed and auto-scaled. PCA, PLS-DA, and heat-mapping analyses were utilized. For hierarchical clustering heatmaps, Euclidean distance measure and Ward clustering method were used. Samples were clustered, independent of each other.

## ACKNOWLEDGMENTS

This research was supported by the National Institutes of Health/NIAID R01-AI148561. The authors acknowledge the BioAnalytical Facility at the University of North Texas for the support with mass spectrometry analyses during this work.

## AUTHOR AFFILIATIONS

[1]Department of Biological Sciences & BioDiscovery Institute, University of North Texas, Denton, Texas, USA
[2]BioAnalytical Facility, University of North Texas, Denton, Texas, USA
[3]Department of Microbiology, The Ohio State University, Columbus, Ohio, USA

## AUTHOR ORCIDs

Adrian Heckart  http://orcid.org/0009-0001-7193-3671
Chad A. Rappleye  http://orcid.org/0000-0001-7880-5958
Ana P. Alonso  http://orcid.org/0000-0003-4696-1811

## FUNDING

| Funder | Grant(s) | Author(s) |
| --- | --- | --- |
| National Institutes of Health | R01-AI148561 | Chad A. Rappleye |
| | | Ana P. Alonso |

## AUTHOR CONTRIBUTIONS

Adrian Heckart, Data curation, Formal analysis, Investigation, Methodology, Validation, Visualization, Writing – original draft | Jean-Christophe Cocuron, Investigation, Methodology, Writing – original draft | Stephanie C. Ray, Investigation, Methodology, Writing – original draft | Chad A. Rappleye, Conceptualization, Funding acquisition, Project administration, Resources, Supervision, Writing – original draft.

## ADDITIONAL FILES

The following material is available online.

### Supplemental Material

**Supplemental Figures (mSystems00186-25-s0001.pdf).** Fig. S1 to S3.
**Table S1, Part 1 (mSystems00186-25-s0002.xlsx).** Table S1a–d.
**Table S1, Part 2 (mSystems00186-25-s0003.xlsx).** Table S1e and f.

**Table S1, Part 3 (mSystems00186-25-s0004.xlsx).** Table S1g–k.

**Table S2 (mSystems00186-25-s0005.xlsx).** Contents of extracellular untargeted metabolomic data sets from MS-DIAL.

**Table S3 (mSystems00186-25-s0006.xlsx).** Comparison of differentially expressed genes in *Histoplasma* with amino acid versus glucose supplementation.

**Table S4 (mSystems00186-25-s0007.xlsx).** Compounds quantified in *Histoplasma* using LC–MS/MS.

**Table S5 (mSystems00186-25-s0008.xlsx).** Electrospray ionization source and MS parameters for negative and positive polarities using SWATH-MS scan survey.

**Table S6 (mSystems00186-25-s0009.xlsx).** MS-DIAL parameters for untargeted metabolomic data processing using GC–MS and LC–MS/MS.

## Open Peer Review

**PEER REVIEW HISTORY (review-history.pdf).** An accounting of the reviewer comments and feedback.

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
