## [Reviewer comments · mSystems]

Revealing pathogenesis-associated metabolites in *Histoplasma capsulatum* through comprehensive metabolic profiling

Adrian Heckart, Jean-Christophe Cocuron, Stephanie Ray, Chad Rappleye, and Ana Alonso

Corresponding Author(s): Ana Alonso, University of North Texas

Review Timeline:

Submission Date:

February 7, 2025

Accepted:

February 11, 2025

Editor: Ákos T. Kovács

Reviewer(s): The reviewers have opted to remain anonymous.

Transaction Report:

DOI: <https://doi.org/10.1128/msystems.00186-25>

Re: mSystems00186-25 (Revealing pathogenesis-associated metabolites in *Histoplasma capsulatum* through comprehensive metabolic profiling)

Dear Prof. Ana Paula Alonso:

Your manuscript has been accepted, and I am forwarding it to the ASM production staff for publication. Your paper will first be checked to make sure all elements meet the technical requirements. ASM staff will contact you if anything needs to be revised before copyediting and production can begin. Otherwise, you will be notified when your proofs are ready to be viewed.

Sincerely,
Ákos T. Kovács
Editor
mSystems

Reviewer #1 (Comments for the Author):

The authors have addressed my concerns.

Reviewer #2 (Comments for the Author):

I sincerely appreciate the thoughtful and rigorous response by the authors. They have thoroughly addressed my comments. I really enjoyed reading this manuscript (again). The work adds important new information to the literature on this understudied by critically important fungal pathogen.